# Modular automated bottom-up proteomic sample preparation for high-throughput applications

Yan Chen[1,2,3], Nurgul Kaplan Lease[1,2,3¤], Jennifer W. Gin[1,2,3], Tadeusz L. Ogorzalek[1,2,3], Paul D. Adams[1,4,5], Nathan J. Hillson[1,2,3], Christopher J. Petzold[1,2,3]*

**1** Joint BioEnergy Institute, Lawrence Berkeley National Laboratory, Emeryville, CA, United States of America, **2** Agile BioFoundry, Lawrence Berkeley National Laboratory, Emeryville, CA, United States of America, **3** Biological Systems & Engineering Division, Lawrence Berkeley National Laboratory, Berkeley, CA, United States of America, **4** Department of Bioengineering, University of California Berkeley, Berkeley, CA, United States of America, **5** Molecular Biophysics and Bioimaging Division, Lawrence Berkeley National Laboratory, Berkeley, CA, United States of America

¤ Current address: Myriad Women's Health, Inc., South San Francisco, CA, United States of America
* cjpetzold@lbl.gov

**Data Availability Statement:** All proteomic data are available via ProteomeXchange with identifier PXD029122 and 10.6019/PXD029122.

## Abstract

Manual proteomic sample preparation methods limit sample throughput and often lead to poor data quality when thousands of samples must be analyzed. Automated liquid handler systems are increasingly used to overcome these issues for many of the sample preparation steps. Here, we detail a step-by-step protocol to prepare samples for bottom-up proteomic analysis for Gram-negative bacterial and fungal cells. The full modular protocol consists of three optimized protocols to: (A) lyse Gram-negative bacteria and fungal cells; (B) quantify the amount of protein extracted; and (C) normalize the amount of protein and set up tryptic digestion. These protocols have been developed to facilitate rapid, low variance sample preparation of hundreds of samples, be easily implemented on widely-available Beckman-Coulter Biomek automated liquid handlers, and allow flexibility for future protocol development. By using this workflow 50 micrograms of protein from 96 samples can be prepared for tryptic digestion in under an hour. We validate these protocols by analyzing 47 *Pseudomonas putida* and *Rhodosporidium toruloides* samples and show that this modular workflow provides robust, reproducible proteomic samples for high-throughput applications. The expected results from these protocols are 94 peptide samples from Gram-negative bacterial and fungal cells prepared for bottom-up quantitative proteomic analysis without the need for desalting column cleanup and with protein relative quantity variance (CV%) below 15%.

## Introduction

Proteomic sample preparation protocols consist of many liquid transfer steps that are well suited for automation with liquid handler systems. As the number of proteomic samples for biotechnological and clinical applications increases, automated solutions will be required to

**Funding:** The funders had and will not have a role in study design, data collection and analysis, decision to publish, or preparation of the manuscript. The United States Government retains and the publisher, by accepting the article for publication, acknowledges that the United States Government retains a non-exclusive, paid-up, irrevocable, worldwide license to publish or reproduce the published form of this manuscript, or allow others to do so, for United States Government purposes. The views and opinions of the authors expressed herein do not necessarily state or reflect those of the United States Government or any agency thereof. Neither the United States Government nor any agency thereof, nor any of their employees, makes any warranty, expressed or implied, or assumes any legal liability or responsibility for the accuracy, completeness, or usefulness of any information, apparatus, product, or process disclosed, or represents that its use would not infringe privately owned rights. The proof-of-concept work and resources were part of the Joint BioEnergy Institute (JBEI; http://www.jbei.org) and extension of the procedure and identification of the sources of error were part of the Agile BioFoundry (ABF; http://agilebiofoundry.org) supported through contract DE-AC02-05CH11231 between Lawrence Berkeley National Laboratory and the U. S. Department of Energy.

**Competing interests:** The authors have declared that no competing interests exist.

minimize human error, save time and resources, and improve the data quality. There have been a number of automated sample preparation protocols developed for both mammalian and bacterial cells that reduce processing time, variability, and overall cost [1–12]. Most of these methods automate the sample cleanup and tryptic digestion portions of the workflow whereas a few automate the entire workflow from cell lysis to digestion [5, 6, 11]. These automation methods show significant improvement in variability and time-savings over manual sample preparation methods. Additionally, high-quality, low variance results can be achieved by researchers without extensive experience in proteomic sample preparation. While automation methods for the full workflow are powerful and convenient they are not as flexible, consequently, when proteomic research projects incorporate new organisms, different amounts of cells, or other variations the entire automated process must be modified. The three protocols described here separate the steps of the fully automated protocol described in Chen et al. [6] to enable flexibility for changing research directions and needs. The modular protocols are much simpler to operate, enable flexible methods development, and process samples in half the time (<1 hour) of the fully-automated protocol due to manual intervention at various steps, such as centrifugation and protein resuspension. Furthermore, the modular automation protocols offer greater flexibility and adaptability without highly-specialized liquid handler systems.

These protocols detail three optimized step-by-step methods to: (A) lyse Gram-negative bacteria and fungal cells; (B) quantify the amount of protein extracted; and (C) normalize the amount of protein and set up tryptic digestion. Importantly, samples prepared through these protocols do not include salts that must be removed prior to LC-MS analysis, thus minimizing sample handling and the associated variance. These protocols have been developed to facilitate rapid, low variance sample preparation of hundreds of samples, be easily implemented on widely-available Beckman-Coulter Biomek automated liquid handlers that use disposable pipet tips, and allow flexibility for future protocol development. By using this modular workflow 96 samples can be prepared for tryptic digestion in under an hour. The tryptic digestion step can be optimized for the given application with many high-throughput digestion protocols such as microwave, elevated temperature, and ultrasonic methods [13, 14] or traditional overnight digestion.

## Materials and methods

The protocol described in this peer-reviewed article is published on protocols.io (dx.doi.org/10.17504/protocols.io.b3gxqjxn) and is included for printing as S1 File with this article." The individual protocols are published on protocols.io (Cell lysis: dx.doi.org/10.17504/protocols.io.b3gsqjwe, Protein quantification: dx.doi.org/10.17504/protocols.io.b3grqjv6, Protein normalization: dx.doi.org/10.17504/protocols.io.b3gtqjwn) and are included for printing as S2–S4 Files with this article.

## Expected results

The modular bottom-up proteomic sample preparation automation protocol (S1 File) is composed of three protocols that detail: (A) cell lysis, protein extraction, protein precipitation; (B) protein quantification; and (C) protein normalization and tryptic digestion. Using the chloroform-methanol protein extraction protocol (S2 File) described, we obtained median amounts of over 115 μg and 50 μg of protein from one OD$^*$mLs (~1 x 10$^9$ cells) of *P. putida* and two OD$^*$mLs *R. toruloides*, respectively (Fig 1). To demonstrate the inter-day variability of the protocol, a single overnight cell culture of *P. putida* and *R. toruloides* was harvested and distribute into two 96 deep well plates and the protocol was repeated on two separate days (Day 1 and Day 7) to demonstrate the reproducibility of the method. The protocol takes 20 minutes to

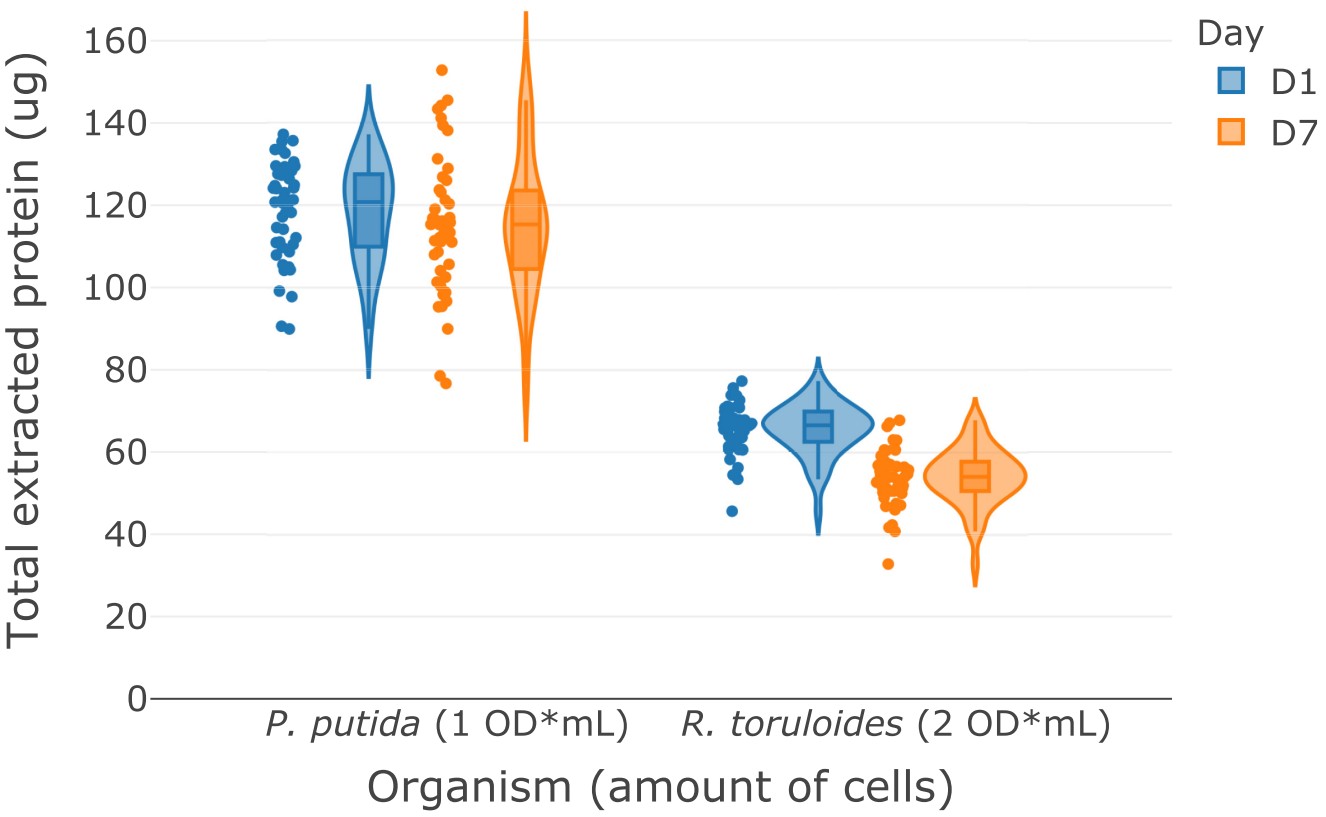

**Fig 1. Violin plots with data points showing the total protein extracted by using the modular automated protocol on *P. putida* and *R. toruloides* from different amounts of biomass (*n* = 47).** D1 and D7 samples correspond to repeat analysis of a single culture of each organism seven days apart to demonstrate the inter-day variability of the protocol.

process one 96-well plate, including centrifugation steps. The amount of protein scaled with the starting amount of biomass which provides flexibility for the desired application. This amount of protein is sufficient for typical nano- and standard-flow LC-MS data acquisition methods and can easily be adjusted for applications requiring larger amounts of protein. The upper limit on the amount of biomass that can be processed with this protocol is limited by the amount of chloroform and methanol that can be added to the PCR plate (~125 μL). For applications that require larger amounts of protein, such as multi-dimensional chromatography, the protocol can easily be adapted to extractions in 96 deep-well plates with more chloroform-methanol. The protocol can also be scaled down to lower cell amounts, but the amount of protein extracted becomes increasingly variable as the amount of cells decreases, so increasing the number of replicates would be advisable. Sample types other than microbial cell pellets, such as tissues and complex biofluids, haven't been tested with this protocol and may need additional preparation steps. Proteins resulting from these samples however are readily suitable for the following two protocols in the workflow.

The protein quantification protocol (S3 File) takes 15 minutes and produces concentration data for two replicates of the samples in a 96-well plate by using the DC protein assay (Bio-Rad), a modified Lowry protein quantification method [15]. The protocol uses a total of 3 μL of each sample and requires aliquoting known concentrations of a BSA standard in a separate plate for calibration curve generation. Duplicate protein quantification was chosen based on

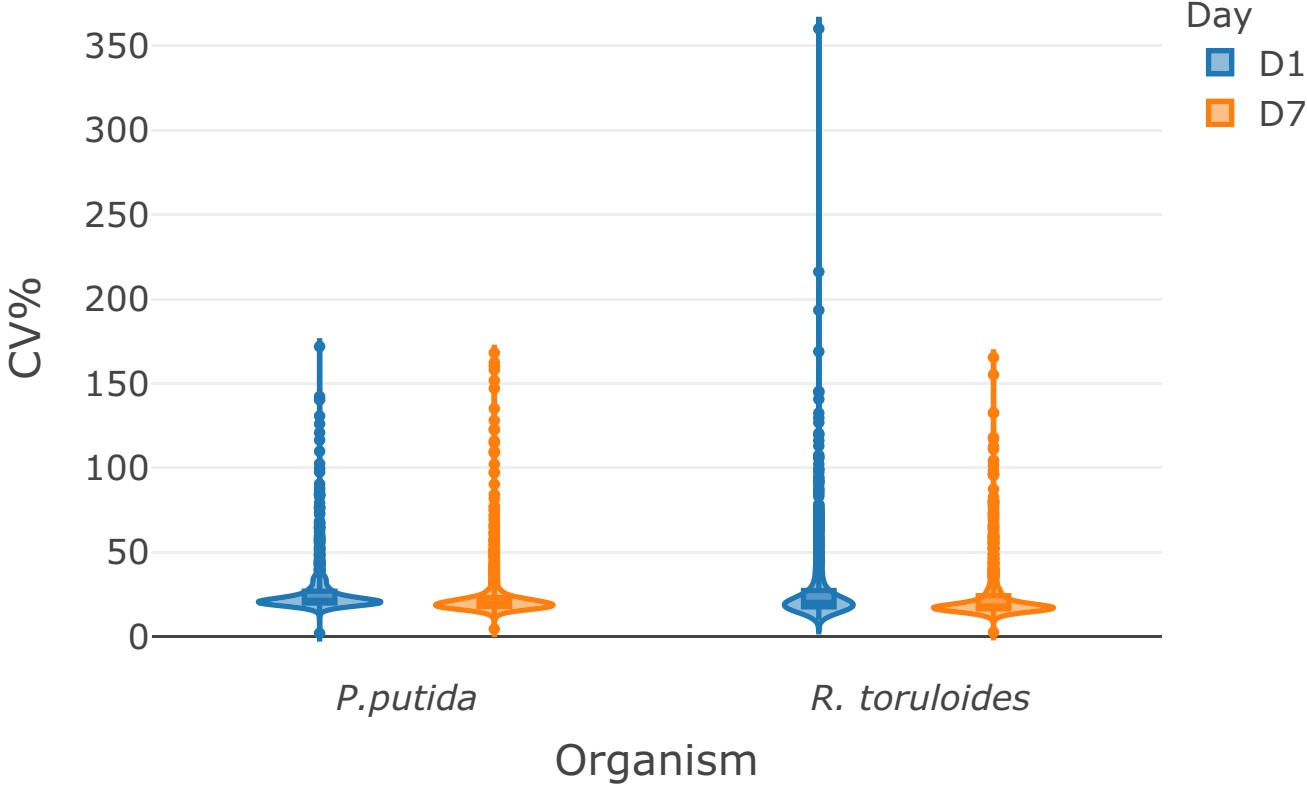

**Fig 2. Reproducibility of the modular automated sample preparation workflow as measured by label-free LC-MS/MS shotgun proteomics analysis.** (A) Violin plots showing the coefficient of variation of MS1 ion intensity quantification for over 900 and 1000 proteins from *P. putida* and *R. toruloides*, respectively (*n* = 47). The violin plots display the kernel density estimation of the CV and inside each violin plot is a box plot summarizing ranges (IQR, whiskers, outlier points) and individual medians (solid lines). The LCMS analysis raw data have been deposited to the ProteomeXchange Consortium data depository at http://www.proteomexchange.org/. They are publicly accessible with the dataset identifier PXD029122 and 10.6019/PXD029122.

previous experience as a balance between sample consumption and accurate concentration measurement. When protein samples are processed by the protein extraction protocol above, the concentration measured by this method falls within a calibration range of 0.125 to 2 µg/µL. For larger or smaller amounts of cells, the concentration may fall outside the calibration range described here, consequently, the dilution factor may need to be adjusted. Once the concentrations of the samples are known the third protocol (S4 File) described here is used to normalize the amount of protein for tryptic digestion and subsequent LC-MS analysis. This protocol takes 20 minutes on the Biomek NX-S8 liquid handler system because the concentration of each well must be adjusted individually. Trypsin, iodoacetic acid, and tris(2-carboxyethyl) phosphine (TCEP) are then added via the Biomek NX-S8 or multi-channel pipette. These protocols are being used for proteomic analysis of metabolically engineered bacteria and fungi. The expected quantitative proteomic results from samples prepared by the modular protocol is demonstrated in Figs 2 and 3 by using an Agilent 1290 UHPLC coupled to a Thermo Orbitrap Exploris 480 system operating in data-dependent acquisition (DDA) mode [16]. The LC-MS/MS method (15 minute total run time) identified over 900 proteins (>6000 peptides) from 14 µg load of *P. putida* protein digest and over 1000 proteins (>4500 peptides) from 10 µg load of *R. toruloides* protein digest. To demonstrate the inter-day variability of the protocol, a single overnight cell culture of *P. putida* and *R. toruloides* was harvested and distribute into two 96 deep well plates at a total cell amount of 1 OD* mL and 2 OD*mL per well, respectively, and

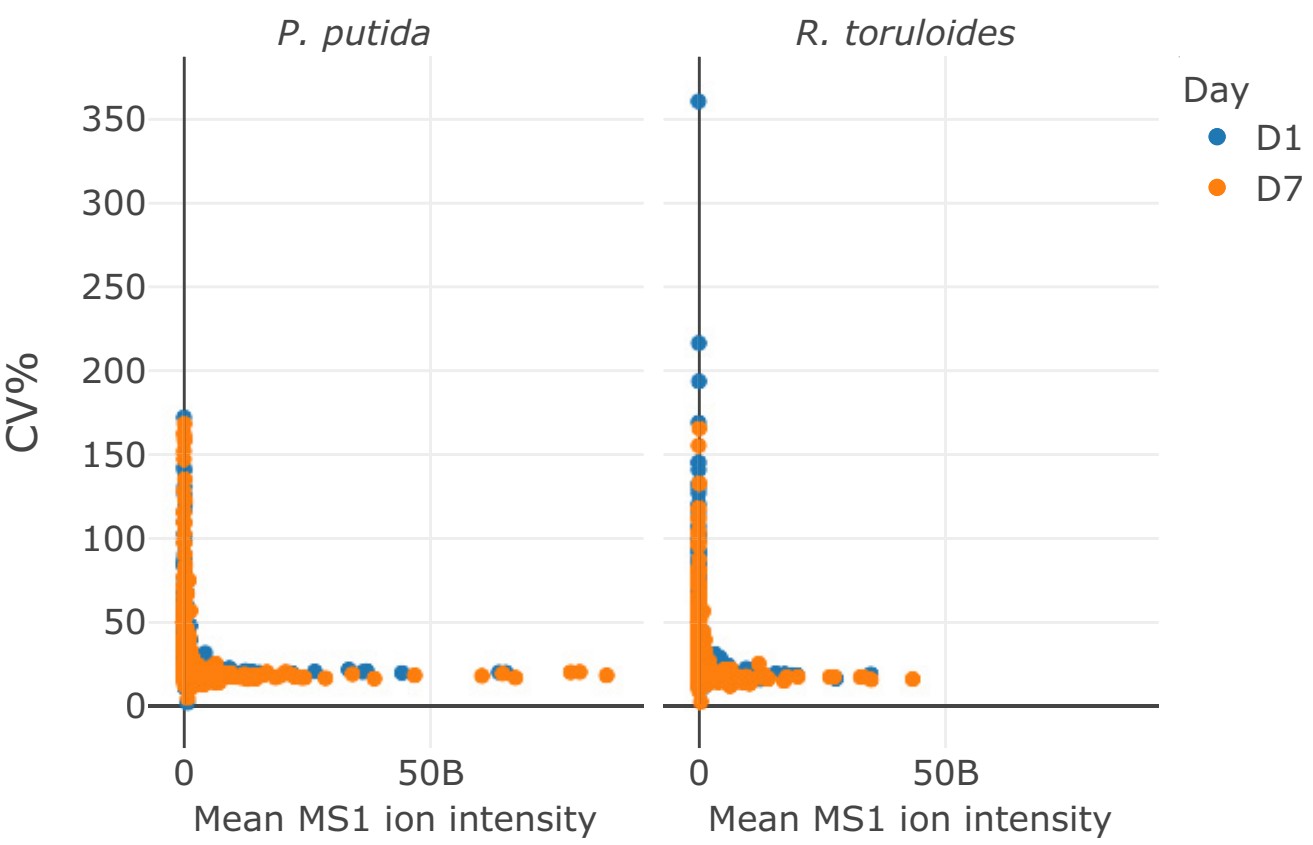

**Fig 3. Scatter plot display of the CV% for each protein (y-axis) vs the mean MS1 ion intensity detected for each protein (x-axis).**

subsequently processed via the modular automation workflow on different days. We used the MS1 ion intensity method with Skyline [17] to quantify over 900 proteins from *P. putida* and over 1000 proteins from *R. toruloides* samples. The median protein variance for the samples were between 18 and 22% CV on two separate days from automated sample preparation protocol (Figs 2 and 3). The high-throughput modular automated protocol enables one researcher to prepare thousands of bottom-up proteomic samples per week. Supporting publications and other organisms are under development.

## Supporting information

**S1 File. Modular automated bottom-up proteomic sample preparation for high-throughput applications.** Also available on protocols.io.
(PDF)

**S2 File. Automated chloroform-methanol protein extraction on the Biomek-FX liquid handler system.** Also available on protocols.io.
(PDF)

**S3 File. Automated protein quantification with the Biomek-FX liquid handler system.** Also available on protocols.io.
(PDF)

**S4 File. Automated protein normalization and tryptic digestion on a Biomek-NX liquid handler system.** Also available on protocols.io.
(PDF)

## Acknowledgments

The authors thank Kristin Burnum-Johnson, Yuzian Gao, and Nathalie Muñoz for helpful discussions about the protocols and Stephen Tan for help with instrumentation.

## Author Contributions

**Conceptualization:** Yan Chen, Christopher J. Petzold.

**Data curation:** Yan Chen, Jennifer W. Gin, Christopher J. Petzold.

**Formal analysis:** Yan Chen, Christopher J. Petzold.

**Investigation:** Yan Chen, Nurgul Kaplan Lease, Jennifer W. Gin, Tadeusz L. Ogorzalek.

**Supervision:** Paul D. Adams, Nathan J. Hillson, Christopher J. Petzold.

**Visualization:** Christopher J. Petzold.

**Writing – original draft:** Yan Chen, Christopher J. Petzold.

**Writing – review & editing:** Yan Chen, Nurgul Kaplan Lease, Jennifer W. Gin, Tadeusz L. Ogorzalek, Paul D. Adams, Nathan J. Hillson, Christopher J. Petzold.

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
