## [Decision Letter · Decision Letter 0]

29 Dec 2021

PONE-D-21-32984Modular automated bottom-up proteomic sample preparation for high-throughput applicationsPLOS ONE

Dear Dr. Petzold,

Thank you for submitting your manuscript to PLOS ONE. After careful consideration, we feel that it has merit but does not fully meet PLOS ONE’s publication criteria as it currently stands. Therefore, we invite you to submit a revised version of the manuscript that addresses the points raised during the review process.

 Review the critiques provided by the reviewers.  This journal ranks clarity, publicly available data, and a high level of detail to support reproducibility of methods.  Keep these criteria in mind as you address the points raised by the reviewers.

We look forward to receiving your revised manuscript.

Kind regards,

John Matthew Koomen, PhD

Academic Editor

PLOS ONE

Journal Requirements:

“The authors have declared that no competing interests exist.”

“The proof-ofconcept work and resources were part of the Joint BioEnergy Institute (JBEI; http://www.jbei.org) and extension of the procedure and identification of the sources of error were part of the Agile BioFoundry (ABF; http://agilebiofoundry.org) supported through contract DE-AC02-05CH11231 between Lawrence Berkeley National Laboratory and the U. S. Department of Energy”

 “The authors have declared that no competing interests exist.”

Reviewers' comments:

Reviewer's Responses to Questions

**Comments to the Author**

1. Does the manuscript report a protocol which is of utility to the research community and adds value to the published literature?

Reviewer #1: Yes

Reviewer #2: Yes

2. Has the protocol been described in sufficient detail?

Descriptions of methods and reagents contained in the step-by-step protocol should be reported in sufficient detail for another researcher to reproduce all experiments and analyses. The protocol should describe the appropriate controls, sample sizes and replication needed to ensure that the data are robust and reproducible.

Reviewer #1: Yes

Reviewer #2: Partly

3. Does the protocol describe a validated method?

Reviewer #1: Yes

Reviewer #2: Yes

4. If the manuscript contains new data, have the authors made this data fully available?

Reviewer #1: No

Reviewer #2: No

**5. Is the article presented in an intelligible fashion and written in standard English?**

Reviewer #1: Yes

Reviewer #2: Yes

6. Review Comments to the Author

Reviewer #1: The authors present an optimized workflow to prepare samples for bottom-up proteomics using automated liquid handling devices. The expected result is less than 15% cv of relative protein abundance.

Specific comments:

1. Data availability statement needs to be edited to reflect accurately where the data has been made available.

2. The authors suggest the method to be a widely applicable proteomics sample preparation workflow, and demonstrate the utility with gram-negative and fungal cell culture. However, many proteomics applications utilize tissue and complex biofluids (serum, lavage fluid, etc) that will likely require significant additional sample preparation steps. The authors should comment on the broader applicability of the method, or revise language to describe suitable sample types

3. The document(s) should be carefully proofread to eliminate typographical errors, specifically with respect to subscripts in chemical formulas.

Reviewer #2: The manuscript by Chen et al. describes a protocol for “automated” sample processing of fungal and bacterial cells for proteomic analysis. Specifically, the authors report 3 separate protocols that are carried out in a commonly used Beckman liquid handler - protein extraction from cells, normalization of protein content, and then trypsin digestion. There is some value in the described approach that could be of interest to a targeted audience; however, there are several issues that need to be addressed prior to publication consideration:

1) The reported protocol seems to be an extension of a previously published research article from their lab, which is within the guidelines of this type of article. However, the authors need to further detail the processing/automation that was mentioned in the first paper but also distinguish any differences. The protocol itself could be more detailed.

2) The authors state that this protocol was very similar to the previously published work but that the updated version was more flexible since it could “operated independently”. This point should be described more clearly and highlighted further in the protocol section if applicable.

3) Some points in the “expected results” don’t match up to the online protocol. For example, the online protocol instructs use of 5ul of sample for protein assay; however, in the results, authors state it uses 3ul of each sample.

4) Description and/or assessment of lysis/lysis efficiency aren’t clear. The main concern is that there might not be adequate representation of the entire proteome (e.g., membrane proteins or other specific classes) or perhaps variability in proteome composition from sample to sample. The authors in the previous paper use MRM to assess specific targets and in this reported protocol show low-coverage, spectral counting data. I understand the potential time savings and benefit of automation, but the authors should expand on limitation of the approach in this regard if applicable.

5) The original paper addresses getting the protein precipitate back in solution by performing “80 cycles of pipetting mixing at the maximum allowable aspiration and dispensing speed” but this is not addressed at all in newer protocol. Based on the extraction procedure used, this seems like a critical point that is important for subsequent processing steps.

6) There is concern of cross contamination between samples and reagents based on the deep well reagent reservoir used (there doesn’t seem to be any separation/ wells in the reservoir). Are there any washes (e.g., needle washes) in between steps?

7) More specifics should be added in the step that describes mixing on deck “with user defined times”. The total step duration to add Methanol shown is 3 minutes – is this mixing only or include other aspects of this overall step?

8) The authors state “In case of large protein pellets, mixing with a multichannel pipette may be necessary”. Is there a particular range you can provide to help the reader assess when that would be required? Related to this, while eliminating most manual pipetting, there are still some manual steps at critical points preventing this from being fully automated.

9) The authors should explain what day 1 and day 7 represent in Figure 1 (how long the organisms were left to grow before collection or days between processing?).

10) In Figure 2, the authors state UHPLC-MRM; however, they describe spectral counting as their quant approach. I believe MRM is copied over from the original paper where they used the MRM approach.

11) Although the mass spectrometry analysis is not necessarily part of the processing protocol, it was used to assess the protocol in terms of protein ID and quant (CV measurement across proteome). A short gradient was used, but the instrument employed should be able to identify more proteins than reported (especially if the protein loading amount on-column reported is correct). A more appropriate quant approach could be used (rather than spectral counting) to determine CVs across the dataset, which could address reproducibility in protein representation across the proteome if the LC-MS system is set up for precise LFQ-based quant. Measurement of the (high-abundance) top few hundred proteins with spectral counting is not really that informative.

7. PLOS authors have the option to publish the peer review history of their article (what does this mean?). If published, this will include your full peer review and any attached files.

Reviewer #1: No

Reviewer #2: No

---

## [Author Response · Author response to Decision Letter 0]

13 Jan 2022

Response to Reviewer Comments:

Reviewer #1: The authors present an optimized workflow to prepare samples for bottom-up proteomics using automated liquid handling devices. The expected result is less than 15% cv of relative protein abundance.

We thank the reviewer for your comments and suggestions. We have responded to your comments below and edited the corresponding text in our manuscript and protocols accordingly. 

Specific comments:

1. Data availability statement needs to be edited to reflect accurately where the data has been made available.

Response: We clarified the text describing where to access the LCMS data in the Data Availability Statement at the end of the manuscript and included it in the caption for Figure 2. The text reads, “The LCMS analysis raw data have been deposited to the ProteomeXchange Consortium data depository at http://www.proteomexchange.org/. They are publicly accessible with the dataset identifier PXD029122 and 10.6019/PXD029122.”

2. The authors suggest the method to be a widely applicable proteomics sample preparation workflow, and demonstrate the utility with gram-negative and fungal cell culture. However, many proteomics applications utilize tissue and complex biofluids (serum, lavage fluid, etc) that will likely require significant additional sample preparation steps. The authors should comment on the broader applicability of the method, or revise language to describe suitable sample types

Response: We clarified the sample types that were used to test the protein extraction protocol and commented on the broader applicability of the protocols established in the workflow in the expected result section. We added text to the Abstract that now reads (new text in bold), “Here, we detail a step-by-step protocol to prepare samples for bottom-up proteomic analysis for Gram-negative bacterial and fungal cells.”

In the Expected Results section, the text now reads, “Sample types other than microbial cell pellets, such as tissues and complex biofluids, haven’t been tested with this protocol and may need additional preparation steps. Proteins resulting from these samples however are readily suitable for the following two protocols in the workflow.”

3. The document(s) should be carefully proofread to eliminate typographical errors, specifically with respect to subscripts in chemical formulas.

Response: Thanks for your comment. We went through all documents and corrected any typo errors we could find.

Reviewer #2: The manuscript by Chen et al. describes a protocol for “automated” sample processing of fungal and bacterial cells for proteomic analysis. Specifically, the authors report 3 separate protocols that are carried out in a commonly used Beckman liquid handler - protein extraction from cells, normalization of protein content, and then trypsin digestion. There is some value in the described approach that could be of interest to a targeted audience; however, there are several issues that need to be addressed prior to publication consideration:

Response: We appreciate your detailed review comments and suggestions for improving our manuscript. We have responded to your comments below and edited the corresponding texts in our manuscript and protocols accordingly.

1) The reported protocol seems to be an extension of a previously published research article from their lab, which is within the guidelines of this type of article. However, the authors need to further detail the processing/automation that was mentioned in the first paper but also distinguish any differences. The protocol itself could be more detailed.

Response: Our manuscript describes a collection of three detailed automation procedures that modularized the full automated proteomic sample preparation process we published previously. The fully automated method requires customized scripts that are difficult to adapt at other labs and requires a highly specialized hybrid liquid handling system with an integrated centrifuge and plate reader which further complicates adoption of the protocol by others. The strengths of the protocols described here are their robustness and application on commonly available liquid handling systems and non-integrated lab equipment. We clarified the text to emphasize the differences between the two methods. 

The text now reads, “While automation methods for the full workflow are powerful and convenient, they are not as flexible, consequently, when proteomic research projects incorporate new organisms, different amounts of cells, or other variations the entire automated process must be modified. The three protocols described here separate the steps of the fully automated workflow described in Chen et al. [6] to enable flexibility for changing research directions and needs. The modular protocols are much simpler to operate, enable flexible methods development, and process samples in half the time (<1 hour) of the fully-automated protocol due to manual intervention at various steps, such as centrifugation and protein resuspension. Furthermore, the modular automation protocols offer greater flexibility and adaptability without highly-specialized liquid handler systems. ”

2) The authors state that this protocol was very similar to the previously published work but that the updated version was more flexible since it could “operated independently”. This point should be described more clearly and highlighted further in the protocol section if applicable.

Response: Thanks for your suggestion. We have added sentences in the introduction section to clarify the “operated independently” point and highlight its advantages in comparison to the fully automated procedure. 

The text now reads, “The modular protocols are much simpler to operate, enable flexible methods development, and process samples in half the time (<1 hour) of the fully-automated protocol due to manual intervention at various steps, such as centrifugation and protein resuspension. Furthermore, the modular automation protocols offer greater flexibility and adaptability without highly-specialized liquid handler systems.”

3) Some points in the “expected results” don’t match up to the online protocol. For example, the online protocol instructs use of 5ul of sample for protein assay; however, in the results, authors state it uses 3ul of each sample.

Response: Thanks for pointing out the error. 5 ul was used in the proof-of-concept protocol version, and later was updated to 3 ul in the method. We updated the protocol to be consistent with the statement in the manuscript. In addition, we moved the Note containing additional information about the dilution to a more prominent place in the Step.

4) Description and/or assessment of lysis/lysis efficiency aren’t clear. The main concern is that there might not be adequate representation of the entire proteome (e.g., membrane proteins or other specific classes) or perhaps variability in proteome composition from sample to sample. The authors in the previous paper use MRM to assess specific targets and in this reported protocol show low-coverage, spectral counting data. I understand the potential time savings and benefit of automation, but the authors should expand on limitation of the approach in this regard if applicable.

Response: The Chloroform-methanol extraction of proteins is a well established method for protein extraction that disrupts lipid membrane and precipitate proteins very effectively without biases to certain classes of proteins. Likewise, the metabolite, protein, and lipid extraction (MPLEx) protocol, has been proven to be robust and applicable to a diverse set of sample types, including cell cultures, microbial communities, and tissues (DOI:https://doi.org/10.1128/mSystems.00043-16). It has proven useful for multi-omics profiling of microbial pathogens (DOI: 10.1039/c6an02486f). Chloroform-methanol extraction method was evaluated to be very effective for the study of membrane proteins of non-model plants (DOI:10.1007/s00425-010-1121-1). We also demonstrated the effectiveness of the chloroform-methanol method in over 50 publications, including using it to accurately quantify membrane electron transport chain proteins in E. coli (DOI: 10.1126/science.aat7925). Furthermore, here, we identified over 900 proteins from 14 μg load of P. putida protein digest and over 1000 proteins from 10 μg load of R. toruloides protein digest with a rapid 15 minutes total LC-MS/MS method, which represent proteins in a wide range of categories including multi-pass integral membrane proteins, cytosolic soluble proteins, lipoproteins, and others. As such, we don’t believe that this is a limitation of the method. 

5) The original paper addresses getting the protein precipitate back in solution by performing “80 cycles of pipetting mixing at the maximum allowable aspiration and dispensing speed” but this is not addressed at all in newer protocol. Based on the extraction procedure used, this seems like a critical point that is important for subsequent processing steps.

Response: The extensive resuspension process in the fully-automated paper was one of the primary issues with process time and reproducibility that we overcame by switching to the modular protocols. We address this issue in the text by adding, “The modular protocols are much simpler to operate, enable flexible methods development, and process samples in half the time (<1 hour) of the fully-automated protocol due to manual intervention at various steps, such as centrifugation and protein resuspension.” 

We highlight this aspect in the protein extraction protocol, step 22 states that the user could define mixing times (an adjustable variable) in the method to mix protein resuspension on deck. The variable mixing time provides users great flexibility to balance method time and efficient protein suspension. We also noted in the same step that users could implement manual pipetting after visual inspection of the resuspension to ensure that no visible protein aggregates in the samples. 

6) There is concern of cross contamination between samples and reagents based on the deep well reagent reservoir used (there doesn’t seem to be any separation/ wells in the reservoir). Are there any washes (e.g., needle washes) in between steps?

Response: The BioMek liquid handling system uses disposable tips instead of fixed tips. New tips were used to address contamination concerns. We added the bold text, shown below, to convey that point. 

“”These protocols have been developed to facilitate rapid, low variance sample preparation of hundreds of samples, be easily implemented on widely-available Beckman-Coulter Biomek automated liquid handlers that use disposable pipet tips, and allow flexibility for future protocol development. 

7) More specifics should be added in the step that describes mixing on deck “with user defined times”. The total step duration to add Methanol shown is 3 minutes – is this mixing only or include other aspects of this overall step?

Response: We specified the mixing step with the number of mixing cycles we used in the demonstration experiments, and clarified that the number of mixing cycles is a variable that users could define themselves as needed. The time shown at each step is an estimated duration of all the components of a step. Specifically, the 3 minutes in your commented step includes tip box moving, tip loading, liquid transfer, and mixing. 

8) The authors state “In case of large protein pellets, mixing with a multichannel pipette may be necessary”. Is there a particular range you can provide to help the reader assess when that would be required? Related to this, while eliminating most manual pipetting, there are still some manual steps at critical points preventing this from being fully automated.

Response: We noted in the step that users need to visually inspect samples after method finishes in case chunks of protein may present in samples of large protein pellets. Our established modular proteomic sample preparation protocols are semi-automatic. 

9) The authors should explain what day 1 and day 7 represent in Figure 1 (how long the organisms were left to grow before collection or days between processing?).

Response: We added text to explain how samples were collected for day 1 and day 7 batches. The text now reads, “To demonstrate the inter-day variability of the protocol, a single overnight cell culture of P. putida and R. toruloides was harvested and distribute into two 96 deep well plates and the protocol was repeated on two separate days (Day 1 and Day 7) to demonstrate the reproducibility of the method.”

We also added “D1 and D7 samples correspond to repeat analysis of a single culture of each organism seven days apart to demonstrate the inter-day variability of the protocol.” to the Figure 1 caption.

10) In Figure 2, the authors state UHPLC-MRM; however, they describe spectral counting as their quant approach. I believe MRM is copied over from the original paper where they used the MRM approach.

Response: Thanks for pointing out the error. We have changed the analysis method to “label-free LC-MS/MS shotgun proteomics analysis”

11) Although the mass spectrometry analysis is not necessarily part of the processing protocol, it was used to assess the protocol in terms of protein ID and quant (CV measurement across proteome). A short gradient was used, but the instrument employed should be able to identify more proteins than reported (especially if the protein loading amount on-column reported is correct). A more appropriate quant approach could be used (rather than spectral counting) to determine CVs across the dataset, which could address reproducibility in protein representation across the proteome if the LC-MS system is set up for precise LFQ-based quant. Measurement of the (high-abundance) top few hundred proteins with spectral counting is not really that informative.

Response: We agree with the reviewer that the Exploris 480 Orbitrap instrument can identify more proteins if longer gradients or nano-flow chromatography are used, but the purpose of the analysis was to demonstrate the use of the protocol for high-throughput applications. As such, the proteome coverage of about 1000 proteins in a microbial sample using a 15 minutes total DDA shotgun method at a standard flow LC-MS system is expected. Users could choose nano-LC systems, longer LC gradients, and DIA workflows to achieve deeper proteome coverage, however, those choices are dependent on the proteomics applications of interest and are separate from the sample preparation protocols detailed here. We chose to increase the number of replicates instead of the depth of proteome coverage to show the power of the sample preparation method for our application. Our data shows high reproducibility for a large number of replicates indicating that this protocol is useful for quantitative proteomic applications. 

Regarding the quantification method, spectral count-based label free quantification (LFQ) is a well-accepted approach that has been applied widely and is discussed in a review by Bantscheff et al (DOI:10.1007/s00216-012-6203-4). And, the Scaffold software used to process the label free quantification is also benchmarked in the paper published in Journal proteomic research (DOI:10.1021/acs.jproteome.6b00645). We agree with the reviewer that other LFQ methods are available and each has its own advantages and disadvantages to be considered. To address the reviewer’s point, we analyzed the data by using the MS1 ion intensity LFQ method in Skyline and observed CVs comparable to that of the spectral count analysis for the data that we use for demonstrating the protocol (updated figures 2 and 3 in the manuscript). We quantified more proteins and still observed a median CV of ~20% for the samples. As such, we replaced Figures 2 and 3 with the results of the MS1 ion intensity quantification analysis. Our data shows that the sample preparation protocols detailed in the manuscript produce data with coefficients of variation that are consistent with good analytical results.

---

## [Decision Letter · Decision Letter 1]

11 Feb 2022

Modular automated bottom-up proteomic sample preparation for high-throughput applications

PONE-D-21-32984R1

Dear Dr. Petzold,

We’re pleased to inform you that your manuscript has been judged scientifically suitable for publication and will be formally accepted for publication once it meets all outstanding technical requirements.

Kind regards,

John Matthew Koomen, PhD

Academic Editor

PLOS ONE

Additional Editor Comments (optional):

Reviewers' comments:

Reviewer's Responses to Questions

**Comments to the Author**

1. Does the manuscript report a protocol which is of utility to the research community and adds value to the published literature?

Reviewer #1: Yes

Reviewer #2: Yes

2. Has the protocol been described in sufficient detail?

Descriptions of methods and reagents contained in the step-by-step protocol should be reported in sufficient detail for another researcher to reproduce all experiments and analyses. The protocol should describe the appropriate controls, sample sizes and replication needed to ensure that the data are robust and reproducible.

Reviewer #1: Yes

Reviewer #2: Yes

3. Does the protocol describe a validated method?

Reviewer #1: Yes

Reviewer #2: Yes

4. If the manuscript contains new data, have the authors made this data fully available?

Reviewer #1: Yes

Reviewer #2: N/A

**5. Is the article presented in an intelligible fashion and written in standard English?**

Reviewer #1: Yes

Reviewer #2: Yes

6. Review Comments to the Author

Reviewer #1: The authors' revised manuscript is acceptable as written. All comments/concerns have been addressed.

Reviewer #2: The authors addressed the comments from my original review.

7. PLOS authors have the option to publish the peer review history of their article (what does this mean?). If published, this will include your full peer review and any attached files.

Reviewer #1: No

Reviewer #2: No

---

## [Editor Report · Acceptance letter]

17 Feb 2022

PONE-D-21-32984R1 

Modular automated bottom-up proteomic sample preparation for high-throughput applications 

Dear Dr. Petzold:

I'm pleased to inform you that your manuscript has been deemed suitable for publication in PLOS ONE. Congratulations! Your manuscript is now with our production department. 

Kind regards, 

on behalf of

Dr. John Matthew Koomen 

Academic Editor

PLOS ONE